# Probing the Promises of Noninvasive Transcranial Electrical Stimulation for Boosting Mental Performance in Sports

**DOI:** 10.3390/brainsci13020282

**Published:** 2023-02-08

**Authors:** Stephane Perrey

**Affiliations:** EuroMov Digital Health in Motion, Univ Montpellier, IMT Mines Ales, 34090 Montpellier, France; stephane.perrey@umontpellier.fr; Tel.: +33-4-3443-2623

**Keywords:** neuromodulation, performance, mental demand, cognitive fatigue, inhibitory control, anxiety, neuroenhancement

## Abstract

While the importance of physical abilities is noncontested to perform in elite sport, more focus has recently been turned toward cognitive processes involved in sport performance. Practicing any sport requires a high demand of cognitive functioning including, but not limited to, decision-making, processing speed, working memory, perceptual processing, motor functioning, and attention. Noninvasive transcranial electrical stimulation (tES) has recently attracted considerable scientific interest due to its ability to modulate brain functioning. Neuromodulation apparently improves cognitive functions engaged in sports performance. This opinion manuscript aimed to reveal that tES is likely an adjunct ergogenic resource for improving cognitive processes, counteracting mental fatigue, and managing anxiety in elite athletes. Nevertheless, the first evidence is insufficient to guarantee its real effectiveness and benefits. All tES techniques could be add-ons to make performance-related cognitive functions more efficient and obtain better results. Modulating inhibitory control through tES over the frontal cortex might largely contribute to the improvement of mental performance. Nevertheless, studies in elite athletes are required to assess the long-term effects of tES application as an ergogenic aid in conjunction with other training methods (e.g., neurofeedback, mental imagery) where cognitive abilities are trainable.

## 1. Introduction

Maximal physical performance has tended to plateau since the 1990s [1]. We can observe that the progress in sport performance of elite athletes has reached more stable levels when compared to the beginning of the 20th century. Achieving in elite competitive sport relies on many factors. Besides its well-recognized effects on muscular and cardiorespiratory functions, physical work produced during exercise is an extraordinary challenge for the brain. With intense and continuous training over years, athletes who reach the elite level are characterized by high physical and motor fitness. A reasonable assumption is that such athletes would also exhibit important adaptations in a large range of cognitive functions. While physical training is the main focus for athletes and conditioning staff, mental performance is also a key component to appraise for performing at the highest level.

Mobilization from a panel of neurocognitive resources is required for an athlete to compete at the highest level. On the whole, sport is a demanding activity requiring more cognitive skills than is often realized. Elite athletes are always looking for innovative ways to improve performance through both physical and cognitive factors, no matter how miniscule that improvement might be. They want to find something extra that allows them to make slight improvements compared to their opponents. The limits of the mind have not yet been fully explored in sports enhancement; however, there exists a range of methods for enhancing various aspects of performance, some of which achieve their effects by directly or indirectly modulating brain activity and/or function. Within this context, some nonpharmacological psychophysiological interventions, such as meditation, autogenic training [2], and biofeedback techniques [3], were proposed to regulate indirectly anxiety and provide optimal sustained attention and inhibitory control to avoid task disengagement, for instance [4]. In neuroscience, it is possible to act more directly due to technological progress reaching the brain organ. Noninvasive brain stimulation methods that include transcranial electric stimulation (tES) and transcranial magnetic stimulation are able to transiently modulate the cortical activity. Nevertheless, the use of such nonpharmacological methods of neuroenhancement is still a regulatory grey area, and raises important issues regarding ethics of sport to achieve a competitive edge [5]. The tES techniques use wearable stimulator delivering weak electrical currents of approximately 0.5 to 2 mA between two (anode and cathode) or more electrodes attached to the scalp, attempting to target specific regions or networks of the brain, making neurons in that configuration more likely to fire. Globally, tES is a neuromodulation technique that produces immediate and lasting excitability changes (i.e., synaptic plasticity). The position of the anode and cathode electrodes on the head is set for inducing current flows to specific brain regions. As a matter of fact, the most used tES forms (transcranial direct current tDCS, transcranial alternating current tACS, and transcranial random noise tRNS stimulation) do not generate action potentials in neurons, but, rather, changes in electrical activity both inside and outside the neurons, leading to modifications in resting membrane potential [6]. The noninvasive neuromodulation techniques have a polarity-dependent inhibitory or excitatory effect on cortical areas of the brain. Particularly, anodal stimulation enhances the firing of neurons by changing the membrane potential while cathodal stimulation inhibits the firing of the neurons by hyperpolarizing them. The tES techniques have been advocated for regarding their effects improving physical performance in sports, even in elite athletes [7], but have also emerged for cognitive performance where the number of peer-reviewed publications remains scarce (see details in Section 3). Collectively, the main targeted brain areas of tES used are the primary motor cortex to improve motor output/physical performance and the dorsolateral prefrontal cortex (dlPFC) to improve memory and cognitive functions. 

In this last decade, there was a great interest for the influence of tES intervention on physical performance, and several systematic reviews were published in the last 5 years [7,8,9,10,11]. Paradoxically, while studies on the influence of tES over dlPFC showed beneficial effect on cognitive performance in patients (e.g., Alzheimer disease [12]), few were focused on boosting cognitive functioning related to sports performance. This is surprising since the cognitive control, for instance, can play a tremendous role for the regulation of fatiguing physical exercise [13], and dlPFC is well known for its role in exerting mental work [14]. In addition, the effect of tDCS on dlPFC improves response inhibition as a key aspect of many sporting skills [15]. Moreover, evidence of the importance of some executive functions (e.g., processing speed, cognitive flexibility, attention, working memory) in sports performance were shown [16,17,18]. Taken together, the effects in individuals with cognitive deficits based on healthy aging or neuropsychiatric diseases appear large, and the current evidence provides an exciting opportunity to use tES to enhance cognitive performance in elite athletes.

Thus, this opinion article aimed at emphasizing the potential major significant impact of tES on the cognitive functions of elite athletes who could make the difference beyond well-trained physical factors. The first section reports how the cognitive functions and brain regions are highly solicited and operating for a number of sports activities. The main well-established papers of this line of research in exercising human subjects were retrieved. Then, the second part takes up a set of major and recent studies highlighting the current promising benefits of exploiting tES techniques in mental performance enhancement. A literature survey was performed to identify supporting studies applying one tES technique to improve cognitive functioning in humans and possibly related sport performance. Based on available studies, three topics (cognitive processes, anxiety/depression, and mental fatigue) were extracted. In elite sport, “brain-boosting” methods for nonpharmacological cognitive enhancement [19] should be considered only as an add-on intervention. What defines elite sport is that it is always able to create room for performance improvement. Within this context, it is important to state that tES as a new technology among others related to sports, for enhancing performance in elite sports (i.e., multipurpose innovation design [20]), was already used in other workplace fields beyond sports. Indeed, in another field requiring performance, the use of tDCS to enhance mental skills of staff members was carried out in research that aims to boost the cognitive performance of air crews, drone operators, and others in the armed forces’ most demanding roles [21].

## 2. Sports Performance Is Cognitively Demanding

There are many sporting situations where the brain plays an important role. First, the brain assumes a key role in determining the volume and/or intensity of exercise an athlete can undertake [22]. Second, the learning as well as execution of technical skills (e.g., swimming, tennis) require a large amount of cognitive demand in coordinating segments of the body, activating muscles at the right time, in the correct sequence, and to an appropriate magnitude. Two types of sport categories are usually proposed. Open-skill sports (e.g., basketball, racket sport) require players to react in an unpredictable and changing environment, while closed-skill sports are performed in a relatively stable and familiar environment in which motor movements follow set patterns (e.g., swimming). Due to these differences, open- and closed-skill sports place various mental demands on athletes [23]. According to Voss et al. [24], athletes from open-skill sports will perform better in some executive functions (e.g., inhibitory control and cognitive flexibility) than those from closed-skill sports. It is of note that many required skills in team sports may be translated to general cognitive domains. A good team player needs excellent executive functions, that is, the cognitive processes that regulate thought and action when novelty occurs. Executive functions refer to a series of interrelated top-down processes needed for behavioral control. We can cite as potential processes involved spatial and divided attention, working memory, problem solving, planning, inhibitory control, and cognitive flexibility. In ball sports, the player must be able to quickly adapt, change strategy by creating new possibilities, and stop behaviors. Cognitive flexibility corresponds to the ability to switch between goals or modes of thinking, such as when switching between tasks or considering multiple alternatives in a complex situation. Indeed, the expert performance approach hypothesis holds that when compared to nonexperts, athletes with a high level of expertise should show superior outcomes across a range of sport-specific cognitive domains (e.g., decision-making, declarative memory, perception, and visual searching capacity) [25]. Finally, athletes frequently encounter high-pressure situations where stress-response regulation can be crucial to performance outcomes. In sport teams, players make decisions with the brain playing a role in analyzing all the stimuli that a player is exposed to, enabling players to make correct decisions while under pressure, with limited time, during various intensities of exercise, and sometimes with limited resources and information [26,27]. 

At the brain level, given its interconnection with other brain areas such as the anterior cingulate cortex and orbitofrontal cortex, premotor cortex, and supplementary motor area, the PFC plays crucial roles in planning, executive processing, and emotional expression. Thus, the PFC is a leading brain area for making spontaneous and self-generated behaviors and for internally driven decision-making. The claim of potential involvement of the PFC in regulating exercise tolerance is supported by the fact that cognitive and emotional factors such as self-motivation impact exercise performance [28]. According to Robertson and Marino [29], the PFC may be involved in regulating and terminating exercise by integrating the afferent feedback from the motor nerves and muscles through the anterior cingulate cortex, the orbitofrontal cortex, or the premotor areas. The brain mechanisms underlying one’s ability to deal with fatigue-related symptoms during exercise will be mainly supported by the PFC and its subregions. More specifically, the frontopolar and dorsolateral regions of the PFC are concerned with inhibitory control, selective attention [30], and the ability to perceive visceral signals (i.e., interoception) and inhibit aversive thoughts derived from exacerbated internal responses [31]. Evidence supports a network of brain regions that are functionally connected with both the anterior cingulate and orbitofrontal cortices during an ongoing cost–benefit analysis [32]. When cost exceeds benefits, PFC regions involved in an inhibitory process underlying cognitive control are disengaged, which quickly leads to the cessation of exercise, as we can find in cyclists or runners. In top soccer players, inhibitory control also occurs inasmuch as it was shown to be associated with successful sporting performance [33]. Interestingly, when combined, mental and physical training will increase cognitive functioning [34]; the two forms of training appear to have an additive effect on neurogenesis.

The activity in the PFC is modulated by the physical strain as well as the cognitive load. In elite sports, the cognitive load is the cognitive demand placed upon athletes as a result of the environmental requirements and task constraints, and the interaction with an individual’s capacity to accommodate such demands [35]. The assessment of cognitive load can be estimated by both the features of the task and the cognitive capacity allocated and required to accommodate the demands imposed by the task [36]. Long periods of highly demanding cognitive activity may increase extracellular brain adenosine concentrations, which in turn may negatively affect rate of perceived exertion and reduce an individual’s ability to make substantial efforts during an exercise [37]. Hence, it is possible that the increased fatigability under stress is related to the temporary cognitive perturbation and diminished PFC activity caused by stress. Accordingly, it appears reasonable to hypothesize that highly demanding cognitive tasks could compromise the ability to deal with the negative thoughts and unpleasant sensations that are naturally evoked during high-intensity exercise sessions [38]. This hypothesis is supported primarily by the fact that the PFC has extensive neuronal connections with other brain regions regulating cognition and emotions [39]. It allows to coordinate brain activity and regulate the catecholamine output. Under stress conditions, the deficient PFC control results in aberrant amygdala activation and related subcortical structures, and deficits in emotion and behavior regulation [40]. Helping athletes or players to control their brain when it matters most: how is that achievable when pressure and mental fatigue are setting in? It is here that tES can modulate this transient brain state. The application of tES on the PFC could strengthen the ability of this region to disregard interoceptive cues (i.e., body signals), keeping the volitional drive to the primary motor cortex and thus delaying the disengagement of the task (i.e., at the end of the exercise).

## 3. Effect of tES on Neurocognitive Performance in Sports

Most tES-based techniques have been shown to promote possible gains in multiple subdomains related to physical performance, such as muscular strength, endurance, and fatigue [7,8,9,10,11,41]. How far can tES improve also mental performance? Theoretically, tES methods are able to favor cognitive performance via acute changes in brain function [42]. If it is proven that tES enhances sports performance, new ethical issues will arise. Currently, tES is not prohibited, but the World Anti-Doping Agency has not yet adopted a strong approach to monitor its use in humans [5,43].

### 3.1. Cognitive Functions

In both healthy and clinical population, plenty of evidence suggests the effectiveness of tES in improving neurocognitive performance in a diverse array of cognitive domains, such as attention [44], working memory [45,46,47,48], reaction time [49], and decision-making [50]. Furthermore, tES could help to manage stress through control of the autonomic nervous system [51]. Overall, these beneficial effects can be useful for improving mental performance in many sports. Currently, there are still few studies testing whether prefrontal cortex stimulation can influence cognitive processes to boost sports performance. Nevertheless, the first findings are promising. They have motivated reflections on studying tES to enhance or modify normal cognitive function, a concept described as a “neuro-doping” intervention [5].

According to stimulation polarity montages, tDCS applied to the left dlPFC and on the right inferior parietal lobe (targeting the default mode network) may reduce the propensity to mind-wander [52], referred to task-unrelated thoughts causing shifts of attention from the task. Anodal tDCS (at 0.5 mA for 19 min over the left DLPFC) was shown to enhance behavioral performance by reducing the interference effect produced by head- fake effect (i.e., interference processing) in basketball players [53]. The most convincing study comes from Borducchi et al. [54]. Following stimulation for ten consecutive weekdays over the left dlPFC (2 mA for 20 min), authors observed improved cognitive performance (tests on memory, attention, and speed of thought) and mood elevation in a sample of elite athletes (judo, swimming, and rhythmic gymnastics). These preliminary evidences indicate that a-tDCS can be used as an advantageous performance-enhancing tool for acute modulation of some neurocognitive functions in athletes. Thus, elite athletes could gain a potential competitive advantage in cognitive performance and mood elevation.

### 3.2. Anxiety and Depression

Competitive sports demand that athletes perform appropriately under intense conditions, not only in the physical but also the psychological context. We all know that many international footballers have missed penalties in FIFA World Cups over the years. Their psychological strength seems to be overridden by the situation when thousands of people are watching in the stadium and millions are watching at home. If there is an imbalance between the ability of the performer and perceived demands of an activity, state anxiety or arousal changes occur. Anxiety has been identified as a very important psychological parameter that influences performance in sport in general [55]. Elite athletes are more vulnerable than the general population to anxiety and depression [56]. Interestingly, Pluhar et al. [57] suggested that individual sport athletes are more likely to exhibit increased anxiety and depression than team sport athletes. 

Prior to a stressful event such as a world class competition, subcortical areas (i.e., hypothalamus, amygdala, and brainstem monoaminergic nuclei) trigger important neuroendocrine responses, notably the activation of the sympathetic adrenal medullary pathway and the hypothalamic–pituitary–adrenal axis. A recent study [58] provided the first preliminary evidence that anodal tDCS (2 mA for 20 min) over the left dlPFC could positively modulate competitive anxiety over the acute stress of competition. Authors suggest that anodal tDCS intervention will favor a top-down regulation of sympathetic adrenal medullary and the hypothalamic–pituitary–adrenal systems. These findings support the notion that tES might be advantageous to enhance sport performance under competitive situations. Due to the high-precision requirements and the related stress, performance in air pistol shooting or biathlon (combining cross-country skiing while carrying a rifle and shooting) can be valuable sport activities for proposing neural priming with anodal tDCS over the dlPFC.

Anxiety and pressure are among the critical causes of the mistakes adversely influencing the quality of a penalty kick. Slutter et al. [59] showed that for players who tended to experience more anxiety and miss penalties, the prefrontal cortex and the left temporal cortex were more active. When experienced players were feeling anxious, their left temporal cortex and PFC activation increased, which could be an indication that experienced players overthink the situation and neglect their skills. Thus, the effectiveness and relevance of tES intervention in a sport-specific context could be investigated more thoroughly by combining EEG-fNIRS-based online monitoring tools. This “neurodiagnostic” framework proposed by Seidel-Marzi and Ragert [60] might be helpful to deliver optimal stimulation parameters from various tES techniques for boosting mental performance during competition. In other domains, improvements in cognitive processing during arithmetic tasks after five consecutive days of tRNS were accompanied by hemodynamic responses consistent with more efficient neurovascular coupling in the stimulated brain regions [61]. The efficacy of tRNS with multielectrode montage for 5 days on learning a complex cognitive task engaging multiple cognitive abilities was recently reported [62].

In the clinical domain, intervention of active tDCS is recognized an effective nonpharmacological treatment for major depressive disorders [63]. In addition, anodal stimulation over the left dlPFC and cathodal stimulation over the right dlPFC appear effective for the treatment of anxiety symptoms in humans [64]. Consequently, we can expect that these types of interventions, while being adapted, could make it possible to face phenomena of acute depressions occurring in elite athletes [56]. Since athletes are constantly expected to perform at the highest level, they are more likely to deal with depression. Based on this, Borducchi et al. [54] proposed that tES could be a valuable tool in helping athletes through elevated mental challenges encountered in elite sport. Interestingly, after anodal tDCS stimulation, they observed that scores on the Beck Depression Inventory scale decreased by 4.5 points, contributing to a greater wellbeing for athletes.

### 3.3. Mental Fatigue

Mental fatigue corresponding to a psychobiological state occurs during and after prolonged high-cognitive-demand activity and can impair selective attention linked to PFC [65]. Among the strategies to counteract mental fatigue, tES techniques might play an important role.

The study of Nikooharf Salehi et al. [66] aimed to examine the effect of tDCS over left dlPFC on mental fatigue and physical performance in fifteen professional swimmers. Subjective ratings of mental fatigue were measured before and after the Stroop task. Their results showed that only anodal tDCS of the left dlPFC reduced negative effects of mental fatigue in 50 m swimming performance, whereas cathodal stimulation had no significant effect. This benefit in fatigue reduction could influence the injury risk profile of an athlete. Recently, Fortes et al. [67] showed that anodal tDCS (2 mA for 30 min) over the orbital PFC was able to maintain endurance performance in mentally fatigued female swimmers. In addition, the same team [68] replicated these findings in male basketball players. They confirmed that anodal tDCS over the motion-sensitive middle temporal area removed the negative effects of mental fatigue on perceptual–cognitive skills (visuomotor and basketball decision-making skills). These performance benefits with anodal tDCS applied over brain frontal areas can be at least attributed to the increased attentional resources and delaying the increased rate of perceived exertion [69,70] due to potential improved inhibitory control. Further, tDCS of the dlPFC appears to be a strong fatigue countermeasure by modulating arousal, and is likely more beneficial than caffeine, for instance [71]. As mentioned before, deciding whether to engage in strenuous mental activities requires trading off the potential benefits against the costs of mental resources. A recent study of Soutshek et al. [72] provided the first insights into the neural mechanisms underlying the willingness to engage in rewarded mental effort. Their findings suggest that some motivational deficits could be treated by tACS over the dorsomedial prefrontal cortex. Altogether, when applied over diverse subregions of PFC, anodal tDCS might improve conjointly executive functions (e.g., inhibitory control, attention), endurance performance, and perception of effort. 

## 4. Conclusions and Future Directions

Enhancement in sport is frequently discussed in the context of enhancement of physical capabilities, but a ceiling effect in athletes’ physical performance may exist at the top level. The first findings presented in this opinion paper suggest that tES (mainly anodal tDCS) intervention add-on can increase cognitive capacities, in turn helping athletes achieve better performance. Specifically, tES has the potential to improve mental performance with regards to heightening/strengthening executive function skills, managing anxiety/depression, and counteracting mental fatigue. Nevertheless, the first evidence is insufficient to guarantee its real effectiveness and benefits. The current findings are based on small sample sizes and low statistical power. Optimal protocols of tES are yet to be clarified in elite athletes to face the potential ceiling effect in performance improvement. Success in competition following improved brain functioning due to tES intervention still needs to be revealed. New studies in elite athletes are required to assess the long-term effects of tES application as an ergogenic aid in conjunction with other training methods (e.g., neurofeedback, mental imagery) where cognitive abilities are trainable. Modulation of brain oscillations in frontal areas from either tACS (exogenous) or EEG neurofeedback (endogenous) are supposed to entrain neuronal networks and in turn substantially enhance the cognitive functions [73]. The various tES techniques provide some new training tools and strategies that remain to be investigated with elite athletes. At the same time, they foster the ethical debate on major issues (e.g., safety, autonomy, accessibility) of using noninvasive transcranial electrical stimulation as a cognitive enhancer in healthy individuals [74] and in the context of athletic performance [43].

## Data Availability

Not applicable.

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
