# Peer review of "Probing the Promises of Noninvasive Transcranial Electrical Stimulation for Boosting Mental Performance in Sports"

_brainsci, 2023, doi:10.3390/brainsci13020282_

Round 1

Reviewer 1 Report

Dear Authors,

I am pleased to review an original manuscript draft entitled "Probing the promises of non-invasive brain stimulation for boosting mental performance in sports" which highlights that transcranial Electrical Simulation can be 10 an adjunct ergogenic resource for improving cognitive processes, counteract mental fatigue and 11 managing anxiety in elite athletes. This paper represents an interesting idea from a quite new angle of view. However, the "opinion" format makes the work a bit deviating from a classic structure, it raises several concerns. Please find my recommendations and suggestions below:

1. Maybe it is a kind of tech issue, but I have not found any references at the end of the manuscript. Could you please provide the references section? This is crucial.

2. Missing the references at the end even aggravates a lack of theoretical framework, which is not sufficiently represented in the paper. It makes a valuable expert opinion soundless. I would invite you to provide a review of at least 3-5 major works in the field. 

3. Due all respect to the format of the paper, a reader should know what theories, approaches and types of logic authors employ in order to achieve results and articulate their opinion. Could you please introduce the methods, design and approach of this paper? Please refer to the literature as well. I would suggest Veal & Darcy and share the file for your convenience.

4. (4) "in the highest level of sport" is vague. Please rephrase it to "elite sport" or "professional sport"... 

5. The objective of the paper is not clearly stated in the abstract and in the introduction as well. Also, the paper's implications, theoretical and practical could be a great extent to this work.

6. (90) "predictable and self-paced" required an explanation.

7. I would suggest introducing the Transcranial Electrical Simulation in greater detail, presenting it as a type of technology with application in sports, please see Glebova & Desbordes, 2021, I share the file for your convenience as well.

8. (153-157) needs further explanation, citation would strengthen the statement.

9. Conclusion could dig deeper, it definitely has potential.

10. Any limitations? Further directions? 

Author Response

Reviewer 1: brainsci-2163423

Author's Reply to the Review Report (Reviewer 1)

I thank the reviewer for her/his thoughtful comments and critiques to the manuscript entitled “Probing the promises of non-invasive brain stimulation for boosting mental performance in sports”. Please find below a point-by-point response to your comments. For clarity, I have repeated your comments in bold and my response is in italics below. Modifications in the manuscript are marked up in red. I appreciate your willingness to consider this revised manuscript for possible publication in Brain Sciences. I would be pleased to provide any further information that you may require.

Dear Authors,

I am pleased to review an original manuscript draft entitled "Probing the promises of non-invasive brain stimulation for boosting mental performance in sports" which highlights that transcranial Electrical Simulation can be 10 an adjunct ergogenic resource for improving cognitive processes, counteract mental fatigue and 11 managing anxiety in elite athletes. This paper represents an interesting idea from a quite new angle of view. However, the "opinion" format makes the work a bit deviating from a classic structure, it raises several concerns. Please find my recommendations and suggestions below:

  1. Maybe it is a kind of tech issue, but I have not found any references at the end of the manuscript. Could you please provide the references section? This is crucial.

Answer: I am very sorry for this tech issue you got from the submission platform. Regarding the first submitted version, PDF and Word files were well uploaded onto the platform (visible on my dash board). Importantly, both files contain 57 references (70 after revision) presenting notably the theoretical framework and the main findings from the current literature covering the topic addressed here in this Opinion paper. I alerted the management Editor regarding this issue that flaws in some way the review (see next comments 2 and 3).

  1. Missing the references at the end even aggravates a lack of theoretical framework, which is not sufficiently represented in the paper. It makes a valuable expert opinion soundless. I would invite you to provide a review of at least 3-5 major works in the field.

Answer: Thank you for giving me the opportunity to add more in the theoretical framework belonging to the introduction section presenting the topic covered and the subsequent section 2 dealing with brain and cognitive functions involved in some sporting contexts; the section 3 being devoted to present the current research works targeted on the tES effects on neurocognitive performance in sports.

I have paid considerable attention to your relevant comments that helped to improve the manuscript overall.Noteworthy that the major works in the field on the tES effects on neurocognitive performance in sports are presented in the dedicated subsections 3.1 (e.g. references 50, 51), 3.2 (e.g. reference 55), 3.3. (e.g. reference 63). In addition, the introduction section was expanded by adding major references related to the theoretical framework (see below).

  1. Due all respect to the format of the paper, a reader should know what theories, approaches and types of logic authors employ in order to achieve results and articulate their opinion. Could you please introduce the methods, design and approach of this paper? Please refer to the literature as well. I would suggest Veal & Darcy and share the file for your convenience.

Answer: Thanks for this suggestion and providing the relevant Book of Veal & Dearcy (2017). I fully agree with the reviewer that some manuscript types, such as original article and systematic review for instance, need some specific sections and dedicated methods ones. Regarding a short opinion manuscript, the usual guidelines for an Opinion article (e.g., https://www.mdpi.com/about/article_types) were followed. Precisely, the present opinion article is corresponding to author’s viewpoints on the interpretation of recent findings (see in section 3), value of the method used supported by evidence and referenced (section 1 . Introduction on tES). The latter points were improved according to your comments. Finally, the current opinion article should be short (usually word count is around 2,000 -3,000 words; first submitted version: 3,061 words vs. revised version: 3,370 words) and focused on the author’s view rather than a comprehensive, critical review. Then, the usual following format was followed: 1) Introduction, 2) Subsections relevant for the subject and discussion 3) Conclusions with future directions.

Altogether, introduction was modified lines 75-86 (relevant literature dealing with the topic, issues), lines 87-91 (objective and presentation of the subsections).

  1. (4) "in the highest level of sport" is vague. Please rephrase it to "elite sport" or "professional sport"...

Answer: thanks for this suggestion. It was changed as proposed: “elite sport”; the “Elite” term being used several times thereafter within the manuscript.

  1. The objective of the paper is not clearly stated in the abstract and in the introduction as well. Also, the paper's implications, theoretical and practical could be a great extent to this work.

Answer: The sentence regarding the objective and the outline of the manuscript was modified in the abstract and at the end of the introduction. Implications and theoretical points are presented throughout the manuscript.

  1. (90) "predictable and self-paced" required an explanation.

Answer: Here, closed skill exercise was redefined for clarity purpose for readers. The sentence (lines 109-110) is now reading as: “Open skill sports (e.g., basketball, racket sport) require players to react in an unpredictable and changing environment while closed skill sports are performed in a relatively stable and familiar environment in which motor movements follow set patterns”.

  1. I would suggest introducing the Transcranial Electrical Simulation in greater detail, presenting it as a type of technology with application in sports, please see Glebova & Desbordes, 2021, I share the file for your convenience as well.

Answer: thanks for this comment and the proposed reference. Transcranial electrical stimulation (tES) was presented lines 44-61 in the first submitted version of the manuscript. This section is dealing with noninvasive brain stimulation methods of which the tES methods. New information was added in this section to complete tES presentation (lines 52-63), although it is an opinion article type (see previous answer).

  1. (153-157) needs further explanation, citation would strengthen the statement.

Answer: Reviewer is correct. The sentence lines 169-173 “This hypothesis is supported primarily by the fact that the PFC exerts an inhibitory control over subcortical regions of the brain such as the amygdala and the hypothalamic-pituitary-adrenal axis” was changed and expanded by adding two key citations.

“This hypothesis is supported primarily by the fact that the PFC has extensive neuronal connections with other brain regions regulating cognition and emotions (Arnsten, 2009). It allows to coordinate brain activity and regulate the catecholamine output. Under stress conditions, the deficient PFC control results in aberrant amygdala activation and related subcortical structures, and deficits in emotion and behavior regulation (Liu et al., 2020).”

  1. Conclusion could dig deeper, it definitely has potential.

Answer: I agree. This last part was modified as suggested. Future directions (Lines 279-281 in the first submitted version) were also expanded in this section.

  1. Any limitations? Further directions?

Answer: Further directions (lines 35-312) and limitations (lines 302-305) are now proposed in the revised conclusion of the manuscript.

Reviewer 2 Report

Thank you to the authors for their time in writing this manuscript and discussing some interesting points surrounding tES in sports. I've described below some points that I feel could be improved in the manuscript.

Abstract: The authors regularly use the term tES and suggest that it could be leveraged to improve mental resources in sports. The authors should consider clarifying non-invasive tES, since it is unlikely the authors are suggesting that people will use invasive tES to enhance sports abilities. 

Introduction: The author again uses tES but mostly appear to be describing tDCS. The author should aim to be more precise in their writing to avoid confusion of describing tDCS when using overarching terms like tES. 

The introduction should also place the premise of the paper in the context of sports enhancement in general. This is obviously a large ongoing discussion surrounding ethics in sports enhancement using substances etc. tES is only a new way of enhancement and should be placed in the context of the larger discussion. 

4th paragraph the authors state:  This"A literature search was performed to identify all relevant studies applying one tES 94 technique to improve cognitive functioning in humans and possibly related sport perfor-95 mance. " -- 

This appears possibly misleading given that the authors say "all relevant studies" which appears subjective. If the authors really searched every paper this would formulate more like a systematic review, or they should discuss their methods used for identifying "all" studies. Otherwise the authors should change this statement to discuss that they surveyed the literature but didn't review "all" papers.

The paper is interesting but given it is an opinion paper it is not entirely novel. The paper would benefit from a discussion on the ethics of using neuromodulation for enhancement of healthy individuals. This discussion should considering addressing the ethics / challenges of risk/reward of doing brain stimulation in healthy individuals. Consider the the dilemma of making brain stimulation available to healthy people when there is already limited access for those with illness, or the fact that some athletes may have access to it and not others. However, you could contrast this with the notion that there are, of course, athletes who may have mental health or neurological conditions such as you describe in terms of depression and anxiety. Or in your section on mental fatigue you could make the point that tES could help to prevent mental fatigue which could help prevent or reduce injuries in sports which is an ethical argument in favor of tES. -- If there are better references on ethics of tES in sport and other discussions surrounding sports enhancement the authors should at minimum reference and point the readers to those papers. 

Do you have a reference for this statement? Is there evidence that athletes experience depression at a higher rate than the public? If so provide a reference, if not the authors should revise: "Since athletes are constantly ex-265 pected to perform at the highest level, they are more likely to deal with depression. " 

3.3. The authors use the term "adverse effects" of mental fatigue. They should revise language as the term "adverse effects" has a very specific meaning in research. 

Author Response

Reviewer 2: brainsci-2163423

Author's Reply to the Review Report (Reviewer 2)

I thank the reviewer for her/his thoughtful comments and critiques to the manuscript entitled “Probing the promises of non-invasive brain stimulation for boosting mental performance in sports”. Please find below a point-by-point response to your comments. For clarity, I have repeated your comments in bold and my response is in italics below. Modifications in the manuscript are marked up in red. I appreciate your willingness to consider this revised manuscript for possible publication in Brain Sciences. I would be pleased to provide any further information that you may require.

Thank you to the authors for their time in writing this manuscript and discussing some interesting points surrounding tES in sports. I've described below some points that I feel could be improved in the manuscript.

Abstract: The authors regularly use the term tES and suggest that it could be leveraged to improve mental resources in sports. The authors should consider clarifying non-invasive tES, since it is unlikely the authors are suggesting that people will use invasive tES to enhance sports abilities.

Answer:  Thanks for this suggestion. “Non-invasive was added in the abstract but also in the title and end of the conclusion.

Introduction: The author again uses tES but mostly appear to be describing tDCS. The author should aim to be more precise in their writing to avoid confusion of describing tDCS when using overarching terms like tES.

Answer: This is correct. Different various non-invasive tES methods were cited in the introduction. Thereafter, the used technique (mainly tDCS or tACS) was proposed within the revised text; tES term was kept for overall considerations and/or used by some manuscripts (review).

The introduction should also place the premise of the paper in the context of sports enhancement in general. This is obviously a large ongoing discussion surrounding ethics in sports enhancement using substances etc. tES is only a new way of enhancement and should be placed in the context of the larger discussion.

Answer:  I agree with you and thanks for this comment. The two first paragraphs of the introduction are presenting the context of sports performance enhancement by highlighting the physical and the mental/mind factors as potential determinants. From lines 44 to 56, some methods (“non-pharmacological psychophysiological interventions”) are proposed, and some new sentences dealing ethics in sports enhancement were added accordingly.

4th paragraph the authors state:  This"literature search was performed to identify all relevant studies applying one tES technique to improve cognitive functioning in humans and possibly related sport performance. " This appears possibly misleading given that the authors say "all relevant studies" which appears subjective. If the authors really searched every paper this would formulate more like a systematic review, or they should discuss their methods used for identifying "all" studies. Otherwise the authors should change this statement to discuss that they surveyed the literature but didn't review "all" papers.

Answer:  Thanks for this helpful comment. I agree that this was misleading as currently stated.  I confirm you that this is not a systematic review. This sentence was changed as “A literature survey was performed to identify supporting studies applying one tES technique to improve cognitive functioning in humans and possibly related sport performance.”

The paper is interesting but given it is an opinion paper it is not entirely novel. The paper would benefit from a discussion on the ethics of using neuromodulation for enhancement of healthy individuals. This discussion should considering addressing the ethics / challenges of risk/reward of doing brain stimulation in healthy individuals. Consider the the dilemma of making brain stimulation available to healthy people when there is already limited access for those with illness, or the fact that some athletes may have access to it and not others. However, you could contrast this with the notion that there are, of course, athletes who may have mental health or neurological conditions such as you describe in terms of depression and anxiety. Or in your section on mental fatigue you could make the point that tES could help to prevent mental fatigue which could help prevent or reduce injuries in sports which is an ethical argument in favor of tES. -- If there are better references on ethics of tES in sport and other discussions surrounding sports enhancement the authors should at minimum reference and point the readers to those papers.

Answer:  Thank you for giving me the opportunity to add more in discussion section regarding ethics of using neuromodulation for enhancement of healthy individuals. This topic was discussed in the following papers:

- Pugh, J., Pugh, C. Neurostimulation, doping, and the spirit of sport. Neuroethics 14 (Suppl 2), 141–158 (2021). https://doi.org/10.1007/s12152-020-09435-7

-Imperatori, L.S., Milbourn, L. & Garasic, M.D. Would the Use of Safe, Cost-Effective tDCS Tackle Rather than Cause Unfairness in Sports? J Cogn Enhanc 2, 377–387 (2018). https://doi.org/10.1007/s41465-018-0113-0

-Voarino N, Dubljevic ́ V and Racine E (2017) tDCS for Memory Enhancement: Analysis of the Speculative Aspects of Ethical Issues. Front. Hum. Neurosci. 10:678. doi: 10.3389/fnhum.2016.00678

As proposed, revised text in the discussion section (and conclusion) presents information addressing the ethics of doing brain stimulation in healthy individuals. The following references were added to point the readers to those papers of interest: Pugh and Pugh (2021) and Imperatori et al. (2017).

Do you have a reference for this statement? Is there evidence that athletes experience depression at a higher rate than the public? If so provide a reference, if not the authors should revise: "Since athletes are constantly expected to perform at the highest level, they are more likely to deal with depression. "

Answer:  This statement was already referenced line 236-237: “Elite athletes are more vulnerable than the general population to anxiety and depression [56].” This reference was added also line 273.

  1. Rice, S.M.; Purcell, R.; De Silva, S.; Mawren, D.; McGorry, P.D.; Parker, A.G.; et al. The mental health of elite athletes: a narrative systematic review. Sports Med. 2016, 46, 1333–1353.

3.3. The authors use the term "adverse effects" of mental fatigue. They should revise language as the term "adverse effects" has a very specific meaning in research.

Answer:  This term was changed. It reads now “negative effects”.

Reviewer 3 Report

Dear authors,

On line 42, I would add this reference after “biofeedback techniques.” (Domingos et al., 2021)

Domingos, C., Silva, C. M. D., Antunes, A., Prazeres, P., Esteves, I., & Rosa, A. C. (2021). The influence of an alpha band neurofeedback training in heart rate variability in athletes. International Journal of Environmental Research and Public Health18(23), 12579.

Furthermore, adding at least one reference when mentioning the previous one (meditation and autogenic) would be good, even if you let them all at the end of the sentence where the actual reference 2 is.

Lines 91 to 94: Be careful with that sentence. What cognitive tasks are we talking about? I’m certain they are not better in ALL cognitive tasks.

Line 130: Missing reference

Author Response

Reviewer 2: brainsci-2163423

Author's Reply to the Review Report (Reviewer 2)

I thank the reviewer for her/his thoughtful comments and critiques to the manuscript entitled “Probing the promises of non-invasive brain stimulation for boosting mental performance in sports”. Please find below a point-by-point response to your comments. For clarity, I have repeated your comments in bold and my response is in italics. Modifications in the manuscript are marked up in red. I appreciate your willingness to consider this revised manuscript for possible publication in Brain Sciences. I would be pleased to provide any further information that you may require.

Comments and Suggestions for Authors

On line 42, I would add this reference after “biofeedback techniques.” (Domingos et al., 2021)

Domingos, C., Silva, C. M. D., Antunes, A., Prazeres, P., Esteves, I., & Rosa, A. C. (2021). The influence of an alpha band neurofeedback training in heart rate variability in athletes. International Journal of Environmental Research and Public Health, 18(23), 12579.

Answer: thanks for this relevant reference (3) that was added in the revised manuscript.

Furthermore, adding at least one reference when mentioning the previous one (meditation and autogenic) would be good, even if you let them all at the end of the sentence where the actual reference 2 is.

Answer: The following refence (2) was added concerning Autogenic training, as suggested.

Litwic-Kaminska, K.; Kotyśko, M.; Pracki, T.; Wiłkość-Dębczyńska, M.; Stankiewicz, B. The Effect of Autogenic Training in a Form of Audio Recording on Sleep Quality and Physiological Stress Reactions of University Athletes—Pilot Study. Int. J. Environ. Res. Public Health 2022, 19, 16043. https://doi.org/10.3390/ijerph192316043

Lines 91 to 94: Be careful with that sentence. What cognitive tasks are we talking about? I’m certain they are not better in ALL cognitive tasks.

Answer: This is a good point. The new sentence is as follows (lines 112-114): According to Voss et al. [22], athletes from open skill sports will perform better in some executive functions (e.g., inhibitory control and cognitive flexibility) than those from closed skill sports.

Line 130: Missing reference

Answer: the sentence L129-132 is now reading as: More specifically, the frontopolar and dorsolateral regions of the PFC are concerned with inhibitory control, selective attention [28], and the ability to perceive visceral signals (i.e., interoception) and inhibit aversive thoughts derived from exacerbated internal responses [29].

The following reference was integrated in the new sentence: [28] C.S. Carter, M. Mintun, J.D. Cohen, Interference and facilitation effects during selective attention: an H215O PET study of Stroop task performance, Neuroimage 2(4) (1995) 264–272.

Round 2

Reviewer 1 Report

Dear Author,

thank you for your efforts to revise the paper, it is a pleasure to see it improved.

However, I would like to clarify a few points from the last round of revisions, hope you will find it helpful.

1. Due all respect to the "opinion" format, to justify an opinion, the author is supposed to mention and at least briefly explain the logic/methods he/she is following in order to have this opinion and achieve conclusions. An opinion cannot simply rise from an unclear pathway. Even if it seems clear to the authors, it should be explained to a reader. Thank you in advance for understanding.

2. (44-60) I would insist on my invitation to introduce tES as a type of sport technology, very briefly explaining what makes it similar and different compared with other types of tech. I share another file for your convenience.

Glebova, E. and Desbordes, M., 2021. Technology innovations in sports: Typology, nature, courses and impact. In Innovation and Entrepreneurship in Sport Management (pp. 57-72). Edward Elgar Publishing.

Author Response

Reviewer 1: brainsci-2163423

Author's Reply to the Review Report (Reviewer 1)

I thank again the reviewer for her/his new comments and critiques to the manuscript entitled “Probing the promises of non-invasive brain stimulation for boosting mental performance in sports”. Please find below a point-by-point response to your last comments. Last modifications in the manuscript are marked up in red. I appreciate your willingness to consider this revised manuscript for possible publication in Brain Sciences. I would be pleased to provide any further information that you may require.

Dear Author,

thank you for your efforts to revise the paper, it is a pleasure to see it improved.

However, I would like to clarify a few points from the last round of revisions, hope you will find it helpful.

  1. Due all respect to the "opinion" format, to justify an opinion, the author is supposed to mention and at least briefly explain the logic/methods he/she is following in order to have this opinion and achieve conclusions. An opinion cannot simply rise from an unclear pathway. Even if it seems clear to the authors, it should be explained to a reader. Thank you in advance for understanding.

Answer. Pathway from the present opinion paper is now presented in the outline at the end of introduction lines 89-97. It was clearly not presented enough in the previous versions. New sentences concerning the logic/methods were added when presenting the two subsections of interest.

  1. (44-0) I would insist on my invitation to introduce tES as a type of sport technology, very briefly explaining what makes it similar and different compared with other types of tech. I share another file for your convenience.

Answer. Reviewer is proposing to introduce tES as type of sport technology. This was done in the revised manuscript lines 100-102 by highlighting that tES may be viewed as a new technology for sports but also for other fields where cognitive performance is paramount. The proposed reference was added in this new sentence.

Glebova, E. and Desbordes, M., 2021. Technology innovations in sports: Typology, nature, courses and impact. In Innovation and Entrepreneurship in Sport Management (pp. 57-72). Edward Elgar Publishing.
